# AUDIOGEN: TEXTUALLY GUIDED AUDIO GENERATION

**Felix Kreuk[1], Gabriel Synnaeve[1], Adam Polyak[1], Uriel Singer[1], Alexandre Défossez[1], Jade Copet[1], Devi Parikh[1], Yaniv Taigman[1], Yossi Adi[1,2]**

[1]FAIR Team, Meta AI
[2]The Hebrew University of Jerusalem
`felixkreuk@meta.com`

## ABSTRACT

We tackle the problem of generating audio samples conditioned on descriptive text captions. In this work, we propose AUDIOGEN, an auto-regressive generative model that generates audio samples conditioned on text inputs. AUDIOGEN operates on a learnt discrete audio representation. The task of text-to-audio generation poses multiple challenges. Due to the way audio travels through a medium, differentiating "objects" can be a difficult task (e.g., separating multiple people simultaneously speaking). This is further complicated by real-world recording conditions (e.g., background noise, reverberation, etc.). Scarce text annotations impose another constraint, limiting the ability to scale models. Finally, modeling high-fidelity audio requires encoding audio at high sampling rate, leading to extremely long sequences. To alleviate the aforementioned challenges we propose an augmentation technique that mixes different audio samples, driving the model to internally learn to separate multiple sources. We curated 10 datasets containing different types of audio and text annotations to handle the scarcity of text-audio data points. For faster inference, we explore the use of multi-stream modeling, allowing the use of shorter sequences while maintaining a similar bitrate and perceptual quality. We apply classifier-free guidance to improve adherence to text. Comparing to the evaluated baselines, AUDIOGEN outperforms over both objective and subjective metrics. Finally, we explore the ability of the proposed method to generate audio continuation conditionally and unconditionally. Samples: `https://felixkreuk.github.io/audiogen`.

## 1 INTRODUCTION

Neural generative models have challenged the way we create digital content. From generating high-quality images (Karras et al., 2019; Park et al., 2019) and speech (Ren et al., 2021; Oord et al., 2016), through generating long textual spans (Brown et al., 2020; Zhang et al., 2022), to the recently proposed text prompted image generation (Ramesh et al., 2022; Rombach et al., 2022), these models have shown impressive results. This begs the question *what would be the audio equivalent to textually guided generative models?* From generating soundscapes to music or speech, a solution to this problem that is high fidelity, controllable, and diverse in its outputs, would be a useful addition to the modern toolbox of creators of movies, video games, and any virtual environments.

While image generation and audio generation have a lot in common, there are a few key differences. Audio is intrinsically a one dimensional signal and thus has less degrees of freedom to differentiate overlapping "objects" (Capon, 1969; Frost, 1972). Real-world audio inherently has reverberations, which makes the task of differentiating objects from the surrounding environment even harder. Moreover, psychoacoustic and psychovisual properties differ, for instance hearing "resolution" (equal-loudness) is U-shaped in frequencies with a dip at 4kHz and bump at 8kHz (Suzuki et al., 2003). Last but not least, the availability of audio data with textual descriptions is orders of magnitude below that of text-image paired data. This makes generating unseen audio compositions a hard task (e.g. generating an audio equivalent of an image of "an astronaut riding a horse in space").

In this work, we tackle the problem of generating audio samples conditioned on descriptive text captions. We additionally extend the proposed method to conditional and unconditional audio continuation. Here, we generate "a dog barks while somebody plays the trumpet in a busy street". In the above prompt, the model must generate three categories of acoustic content, with varying degrees of background/foreground, durations, and relative position in the temporal axis, the composition of which is highly unlikely to be present in the training set. Generating such audio is thus a challenge in generalization, acoustic fidelity, production and mastering.

We propose AUDIOGEN, an autoregressive textually guided audio generation model. AUDIO-GEN consists of two main stages. The first encodes raw audio to a discrete sequence of tokens using a neural audio compression model (e.g. Zeghidour et al. (2021)). This model is trained in an end-to-end fashion to reconstruct the input audio from the compressed representation, with an addition of a perceptual loss in the form of a set of discriminators. Such an audio representation is designed to generate high-fidelity audio samples while still being compact. The second stage, leverages an autoregressive Transformer-decoder language-model that operates on the discrete audio tokens obtained from the first stage while also being conditioned on textual inputs. We represent text using a separate text encoder model pre-trained on a large corpus of text, namely T5 (Raffel et al., 2020). The pre-trained text encoder enables the generalization to text concepts that are absent from current text-audio datasets. This is especially important when working with text annotations limited in terms of diversity and descriptiveness.

Compared to the existing text-to-audio work (Yang et al., 2022), AUDIOGEN generates samples that obtain better objective and subjective metrics. In particular, AUDIOGEN creates more natural sounding unseen audio compositions. Lastly, we empirically show how the proposed approach can be extended to audio continuation considering both conditional and unconditional generation.

**Our contributions:** (i) We propose a state-of-the-art auto-regressive audio generation model conditioned on textual descriptions or audio prompts, as evaluated with objective and subjective (human listeners) scores. Specifically we propose two model variations, one with 285M parameters and another one with 1B parameters; (ii) We improve text-to-audio generation in two axes. We improve text adherence by applying classifier free guidance on top of the audio language model. We improve compositionality by performing on the fly text and audio mixing; (iii) We show that the proposed approach can be extended to audio continuation conditioned on text and unconditionally; (iv) We explore the trade-off between audio-fidelity and sampling time by utilizing residual vector quantization (for acoustic units) and multi-stream transformers.

## 2 RELATED WORK

**Speech Representation Learning.** Studies on unsupervised speech representation learning can be roughly divided into reconstruction and self-supervised learning methods. Auto-encoding is the common approach for signal reconstruction, where speech is first encoded into a low-dimensional latent representation, and then decoded back to speech. Various constraints can be imposed on the encoded space, such as temporal smoothness (Ebbers et al., 2017), discreteness (van den Oord et al., 2017b), and hierarchy (Hsu et al., 2017). Self-Supervised Learning (SSL) methods for speech have shown remarkable results for automatic speech recognition (Schneider et al., 2019; Baevski et al., 2020; Wang et al., 2021), phoneme segmentation (Kreuk et al., 2020), and audio compression (Zeghidour et al., 2021; Polyak et al., 2021). van den Oord et al. (2018) and Schneider et al. (2019) suggested training a convolutional neural network to distinguish true future samples from random distractor samples using a Contrastive Predictive Coding (CPC) loss function. Ao et al. (2022) proposed a speech version of the T5 model and showed its efficiency on various speech tasks. Similar to CPC, Baevski et al. (2020) use an encoder and a predictor, which is trained contrastively to distinguish positive and negative samples. Unlike (Schneider et al., 2019), it discretizes and masks segments of the encoder's output. Hsu et al. (2021) proposed the HuBERT model which is trained with a masked prediction task similar to BERT (Devlin et al., 2019) but with masked continuous audio signals. Chen et al. (2022) proposed a similar version of HuBERT trained on larger and augmented dataset. More recently, Huang et al. (2022) proposed a Masked Auto Encoding approach for learning a speech representation and show it efficiency on several audio classification tasks.

Another line of relevant prior work relates to modeling audio discrete representations. Recent studies suggest quantizing SSL representations using k-means and later perform language modeling (Lakho-

tia et al., 2021; Kharitonov et al., 2022a; Borsos et al., 2022), multi-stream processing (Kharitonov et al., 2022b), speech emotion conversion (Kreuk et al., 2022), spoken dialogue (Nguyen et al., 2022), and speech-to-speech translation (Lee et al., 2022a;b; Popuri et al., 2022).

**Text-to-Image** has seen great advances recently. DALL-E (Reddy et al., 2021) first transforms the patches of an image to discrete codes using a pre-trained VQ-VAE. During training, codes representing image patches were appended to codes representing text. Then, a Transformer-decoder model was trained to model these codes in an autoregressive fashion, while Gafni et al. (2022) suggested a similar approach and incorporated segmentation maps for increased controllability. In the Parti model (Yu et al.) the authors suggested modeling the task of text-to-image as a sequence-to-sequence problem using Transformers in an encoder-decoder architecture.

More recently, diffusion model have gained increased popularity (Nichol et al., 2022; Ramesh et al., 2022; Saharia et al.; Rombach et al., 2022). DALLE-2 (Ramesh et al., 2022) used a diffusion model to predict the CLIP visual features given the CLIP text encoding (prior), and another diffusion model to predict the image pixels given the predicted CLIP visual features (decoder). The predicted image is upsampled to a higher resolution using a cascade of super-resolution models. Imagen (Saharia et al.) employed a similar approach but omitted the prior component in favor of using pre-trained text-encoders such as T5 (Raffel et al., 2020).

**Text-to-Audio.** The most relevant to our work is the one proposed by Yang et al. (2022), in which the authors proposed DiffSound, a text-to-audio model based on a diffusion process that operates on audio discrete codes. The audio codes were obtained from a VQ-VAE (van den Oord et al., 2017a) based model trained over mel-spectrogram. To further improve model performance, Yang et al. (2022) suggested pre-training the diffusion model using labeled tags with a random input masking. They additionally explore the usage of an auto-regressive Transformer decoder model, however found it to be inferior to the diffusion based model.

The proposed method differentiate from DiffSound in the following: (i) our audio representation is being learned directly from the raw-waveform; (ii) we create new audio compositions using data augmentation allowing the model to generate audio from complex text captions; (iii) we apply and study the effect of classifier free guidance under the auto-regressive setting; (iv) in contrast to Yang et al. (2022), we empirically demonstrate that text-conditioned auto-regressive models can generate high-quality audio samples.

## 3 METHOD

The proposed method, AUDIOGEN, is based on two main steps: (i) learning discrete representation of the raw audio using an auto-encoding method; (ii) training a Transformer language model over the learnt codes obtained from the audio encoder, conditioned on textual features. Then, during inference time, we sample from the language model to generate a new set of audio tokens given text features. These tokens can later be decoded into the waveform domain using the decoder component from step (i). A visual description of the proposed method can be seen on Figure 1.

### 3.1 AUDIO REPRESENTATION

An audio signal of duration $d$ can be represented by a sequence $x \in [-1, 1]^{C_a \times T}$ with $C_a$ the number of audio channels, $T = d \cdot f_{sr}$ the number of audio samples at a given sample rate $f_{sr}$. In this work we set $f_{sr} = 16$kHz. The audio representation model is composed of three components: (i) an encoder network $E$ which gets as input an audio segment and outputs a latent representation $z$; (ii) a quantization layer $Q$ produces a compressed representation $z_q$, using a Vector Quantization (Vasuki & Vanathi, 2006) layer; (iii) a decoder network $G$ reconstructs the time-domain signal, $\hat{x}$, from the compressed latent representation $z_q$. The whole system is trained end-to-end to minimize a reconstruction loss applied over both time and frequency domain, together with a perceptual loss in the form of several discriminators operating at different temporal resolutions. Using a pre-trained model, we can leverage the encoder and quantizer components as a discrete feature extractor (i.e., $Q \circ E$) and $G$ to decode the representation to the time-domain signal. For $Q$, we use a single codebook with 2048 codes, where each code is a 128 dimensional vector. A visual description of the proposed method can be seen in Figure 1 (right).

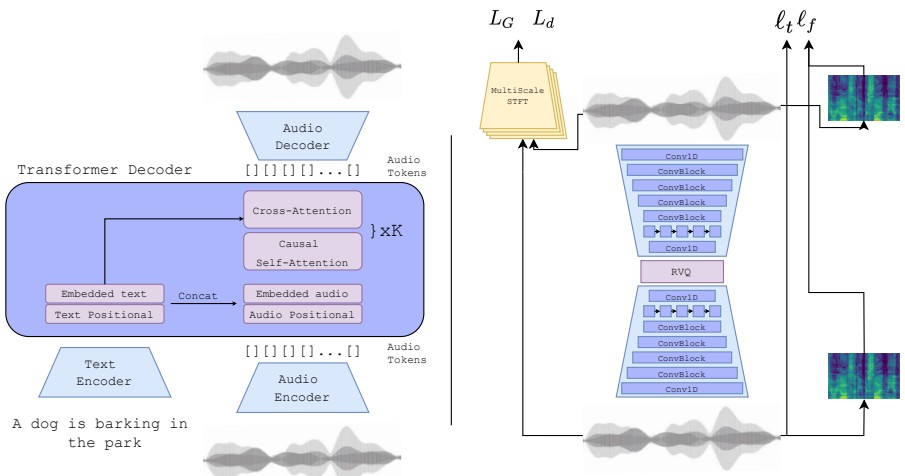

Figure 1: A general overview of the AUDIOGENsystem. Left: the audio representation model. Right: the audio language model. Both text and audio embeddings are concatenated over the time dimension and fed in K causal self-attention and cross-attention blocks with the embedded text.

**Architecture.** We follow a similar auto-encoder model architecture as in Zeghidour et al. (2021); Li et al. (2021). The encoder model $E$ consists of a 1D convolution with $C$ channels followed by $B$ convolutional blocks. Each convolutional block is composed of a single residual unit followed by a down-sampling layer consisting of a strided convolution, with a kernel size $K$ of twice the stride $S$. The residual unit contains two convolutions and a skip-connection. The number of channels is doubled whenever down-sampling occurs. The convolutional blocks are followed by a two-layer LSTM for sequence modeling and a final 1D convolution layer with a kernel size of 7 and $D$ output channels. We use $C = 32$, $B = 4$ and $(2, 2, 2, 4)$ as strides. We use ELU as a non-linear activation function (Clevert et al., 2015) and a LayerNorm (Ba et al., 2016). The decoder mirrors the encoder, using transposed convolutions instead of strided convolutions, and with the strides in reverse order as in the encoder, outputting the final audio.

**Training Objective.** We optimize a GAN based training objective similar to (Kong et al., 2020; Zeghidour et al., 2021) of jointly minimizing a combination of reconstruction losses and adversarial losses. Specifically, we minimize the L1 distance between the target and reconstructed audio over the time domain, i.e. $\ell_t(\boldsymbol{x}, \hat{\boldsymbol{x}}) = \|\boldsymbol{x} - \hat{\boldsymbol{x}}\|_1$. For the frequency domain loss, we use a linear combination between the L1 and L2 losses over the mel-spectrogram using several time scales (Yamamoto et al., 2020; Gritsenko et al., 2020). Formally,

$$\ell_f(\boldsymbol{x}, \hat{\boldsymbol{x}}) = \frac{1}{|\alpha| \cdot |s|} \sum_{\alpha_i \in \alpha} \sum_{i \in e} \|\mathcal{S}_i(\boldsymbol{x}) - \mathcal{S}_i(\hat{\boldsymbol{x}})\|_1 + \alpha_i \|\mathcal{S}_i(\boldsymbol{x}) - \mathcal{S}_i(\hat{\boldsymbol{x}})\|_2, \tag{1}$$

where $\mathcal{S}_i$ is a 64-bins mel-spectrogram using a normalized STFT with window size of $2^i$ and hop length of $2^i/4$, $e = 5, \ldots, 11$ is the set of scales, and $\alpha$ represents the set of scalar coefficients balancing between the L1 and L2 terms. Unlike Gritsenko et al. (2020), we set $\alpha_i = 1$.

To further improve the quality of the generated samples, we additionally optimize a multi-scale STFT-based (MS-STFT) discriminator. Multi-scale discriminators are popular for capturing different structures in audio signals (Kumar et al., 2019; Kong et al., 2020; You et al., 2021). The MS-STFT discriminator is based on identically structured networks operating on multi-scaled complex-valued STFT where its real and imaginary parts are concatenated. Each sub-network is composed of a 2D convolutional layer (using kernel size 3 x 8 with 32 channels), followed by 2D convolutions with increasing dilation rates in the time dimension of 1, 2 and 4, and a stride of 2 over the frequency axis. A final 2D convolution with kernel size 3 x 3 and stride (1, 1) provide the final prediction. We use 5 different scales with STFT window lengths of [2048, 1024, 512, 256, 128]. The adversarial loss for the generator is constructed as follows, $\ell_g(\hat{\boldsymbol{x}}) = \frac{1}{K} \sum_k \max(0, 1 - D_k(\hat{\boldsymbol{x}}))$, where $K$ is the number of discriminator networks. Similarly to previous work on neural vocoders (Kumar et al., 2019; Kong et al., 2020; You et al., 2021), we additionally include a feature matching loss for the

generator. Formally,

$$\ell_{feat}(\boldsymbol{x}, \hat{\boldsymbol{x}}) = \frac{1}{KL} \sum_{k=1}^{K} \sum_{l=1}^{L} \|D_k^l(\boldsymbol{x}) - D_k^l(\hat{\boldsymbol{x}})\|_1, \tag{2}$$

where $(D_k)$ are the discriminators, and $L$ is the number of layers in discriminators.

Overall the discriminators are trained to minimize the following: $L_d(\boldsymbol{x}, \hat{\boldsymbol{x}}) = \frac{1}{K} \sum_{k=1}^{K} \max(0, 1 - D_k(\boldsymbol{x})) + \max(0, 1 + D_k(\hat{\boldsymbol{x}}))$, where $K$ is the number of discriminators, while the generator is trained to minimize the following: $L_G = \lambda_t \cdot \ell_t(\boldsymbol{x}, \hat{\boldsymbol{x}}) + \lambda_f \cdot \ell_f(\boldsymbol{x}, \hat{\boldsymbol{x}}) + \lambda_g \cdot \ell_g(\hat{\boldsymbol{x}}) + \lambda_{feat} \cdot \ell_{feat}(\boldsymbol{x}, \hat{\boldsymbol{x}})$.

## 3.2 Audio Language Modeling

Recall, in this work our goal is to generate audio conditioned on text. Specifically, given a textual input $\boldsymbol{c}$ the Audio Language Model (ALM) component outputs a sequence of audio tokens $\hat{\boldsymbol{z}}_q$, which can be later decoded into raw audio using $G$.

Consider a text encoder $F$ which maps a raw text input into a semantic dense representation, $F(\boldsymbol{c}) = \boldsymbol{u}$. Then, a Look-Up-Table (LUT) embeds the audio tokens, $\hat{\boldsymbol{z}}_q$, into a continuous space, $\text{LUT}(\hat{\boldsymbol{z}}_q) = \boldsymbol{v}$. We then concatenate both $\boldsymbol{u}$ and $\boldsymbol{v}$ to create $Z = \boldsymbol{u}_1, \dots, \boldsymbol{u}_{T_u}, \boldsymbol{v}_1, \dots, \boldsymbol{v}_{T_v}$, where $T_u$ and $T_v$ are the length of the text representation and audio representation respectively.

Using the above representation, we train a Transformer-decoder language-model parameterized by $\theta$ using the cross-entropy loss function:

$$L_{\text{LM}} = - \sum_{i=1}^{N} \sum_{j=1}^{T_v} \log p_\theta(\boldsymbol{v}_j^i | \boldsymbol{u}_1^i, \dots, \boldsymbol{u}_{T_u}^i, \boldsymbol{v}_1^i, \dots, \boldsymbol{v}_{j-1}^i). \tag{3}$$

The text representation is obtained using a pre-trained T5 text-encoder (Raffel et al., 2020). We additionally experimented with learning text embeddings using a LUT. Although it produces comparable results to the T5 model, it limits our ability to generalize to unseen words during training, hence we did not pursue this direction. The transformer-decoder language-model is implemented using a GPT2-like architecture (Radford et al., 2019). To achieve a better text adherence we add cross-attention between audio and text to each attention block of the transformer. See a visual description on Figure 1 (left). The entire system may be alternatively viewed as an encoder-decoder model, where the encoder (T5) is pre-trained and fixed throughout training.

**Classifier Free-Guidance.** It was recently shown by Ho & Salimans (2021); Nichol et al. (2022) that using the Classifier Free Guidance (CFG) method is an effective mechanism for controlling the trade-off between sample quality and diversity. Although the CFG method was originally proposed for the score function estimates of diffusion models, in this work we apply it to auto-regressive models. During training we optimize the Transformer-LM conditionally and unconditionally. In practice, we randomly omit the text conditioning in 10% of training samples. At inference time we sample from a distribution obtained by a linear combination of the conditional and unconditional probabilities. Formally we sample from,

$$\gamma \log p_\theta(\boldsymbol{v}_j^i | \boldsymbol{u}_1^i, \dots, \boldsymbol{u}_{T_u}^i, \boldsymbol{v}_1^i, \dots, \boldsymbol{v}_{j-1}^i) + (1 - \gamma) \log p_\theta(\boldsymbol{v}_j^i | \boldsymbol{v}_1^i, \dots, \boldsymbol{v}_{j-1}^i), \tag{4}$$

where $\gamma$ is the guidance scale.

**Multi-stream audio inputs.** In order to generate high-quality audio samples we down-sample the raw audio by a factor of 32, which corresponds to 2 ms for each audio token. This requires us to operate over extremely long sequences as each second of audio is represented by 500 tokens. Modeling such long sequences is a notoriously difficult problem (Rae et al., 2020; Zaheer et al., 2020; Beltagy et al., 2020). To alleviate this problem, we propose a *Multi-Stream* representation and modeling paradigm. It was shown by Kharitonov et al. (2022b) that transformers are capable of modeling multiple streams simultaneously.

Consider a sequence of length $T_v$, we can learn a representation of length $T_v/2$ using two parallel streams of approximately the same bit-rate. This approach can be generalized to $k$ streams, where each stream is of length $T_v/k$ and each codebook is of size $2048/k$. Such representation can be obtained by generalizing $Q$ from a single code book Vector-Quantization to a Residual Vector

Quantization module as done by Zeghidour et al. (2021). At time $t$, the network is fed with $k$ discrete codes, which then embedded using $k$ embedding layers. The final embedding at time $t$ is the mean of these $k$ embeddings. We adapt the network to output $k$ codes using $k$ LM prediction heads. The prediction heads operate independently of each other, we explored conditioning stream $i$ on stream $i - 1$ but did not observe any performance gains.

## 4 EXPERIMENTS

In this section we start by providing a detailed description of the experimental setup. Next, we present the main results for audio generation, and we conclude this section with an ablation study.

### 4.1 EXPERIMENTAL SETUP

**Dataset.** We use a set of several datasets: AudioSet (Gemmeke et al., 2017), BBC sound effects [1], AudioCaps (Kim et al., 2019), Clotho v2 (Drossos et al., 2020), VGG-Sound (Chen et al., 2020), FSD50K (Fonseca et al., 2021), Free To Use Sounds [2], Sonniss Game Effects [3], WeSoundEffects [4], Paramount Motion - Odeon Cinematic Sound Effects [5]. All audio files were sampled at 16kHz.

For textual descriptions we use two types of annotations. The first one is multi-label annotations, available for the datasets: AudioSet, VGG-Sound, FSD50K, Sinniss Game Effects, WeSoundEffects, Paramount Motion - Odeon Cinematic Sound Effects. We form pseudo-sentences by concatenating lists of tags available per audio samples (e.g., "dog, bark, park" is transformed to "dog bark park"). The second type of annotation is natural language captions available for the datasets: AudioCaps, Clotho v2, Free To Use Sounds, and BBC Sound Effects. A more elaborate description of the used datasets can be found in Appendix 3. We apply a pre-processing step to better match the class-label annotation distribution. Specifically, we remove stop words and numbers, finally we lemmatize the remaining words (e.g. "a dog is barking at the park" is transformed to "dog bark park") using the WordNet lemmatizer in NLTK (Bird et al., 2009). As speech is the dominant class in the data, we filter all samples where the tag or caption contains the word "speech" to generate a more balanced dataset. Overall we are left with ~4k hours for training data.

**Data Augmentations.** One of the most impressive capabilities of recently proposed generative models (Ramesh et al., 2022; Saharia et al.; Gafni et al., 2022) is their ability to create unseen object compositions (e.g., "An astronaut riding a horse in space"). To achieve similar capabilities with regards to audio generation we propose an augmentation method that fuses pairs of audio samples and their respective text captions, thus creating new concept compositions during training. Formally, given two audio samples $x_1, x_2$ and their respective text captions $c_1, c_2$, we first randomly draw an temporal offset to merge the two audio samples. Next, we draw a random Signal-to-Noise Ratio (SNR) in the interval $[-5, 5]$, and finally we mix the audio samples and concatenate the text captions $c_1, c_2$.

**Evaluation Methods.** We evaluate all models and baselines using both objective and subjective metrics. For the objective functions we compute the Fréchet Audio Distance (FAD) (Kilgour et al., 2019) over both real and generated samples. FAD, adapted from the Fréchet Inception Distance (FID) to the audio domain, is a reference-free evaluation metric that closely correlates with human perception. Similarly to Yang et al. (2022) we additionally compute the KL-Divergence between the output of a state-of-the-art audio classification model (Koutini et al., 2021) while feeding it both the original samples and the generated audio. The FAD was shown to correlate well with human perception in terms of audio quality. On the other hand, the KL measure is computed using the label distribution produced by a pre-trained classifier. Hence, it reflects more on the broader audio concepts occurring in the recording. As a result, the two metrics are complementary.

For subjective methods, we follow the a similar setting to (Yang et al., 2022). We ask human raters to evaluate two main aspects of the audio signal (i) overall quality (OVL), and (ii) relevance to the

---

[1] https://sound-effects.bbcrewind.co.uk/

[2] https://www.freetousesounds.com/all-in-one-bundle/

[3] https://sonniss.com/gameaudiogdc

[4] https://wesoundeffects.com/we-sound-effects-bundle-2020/

[5] https://www.paramountmotion.com/odeon-sound-effects

Table 1: Results are reported for DiffSound together with several versions of AUDIOGEN. For DiffSound data augmentation, we follow the authors suggested mask-based text generation (MBTG) strategy. For subjective tests we report overall quality (OVL), and text relevenace (REL.) together with 95% Confidence Interval. For the objective metrics we report FAD and KL.

| | #params | AUG. | TEXT-COND. | SUBJECTIVE | | OBJECTIVE | |
| --- | --- | --- | --- | --- | --- | --- | --- |
| | | | | OVL↑ | REL.↑ | FAD↓ | KL↓ |
| Reference | - | - | - | 92.08±1.16 | 92.97±0.85 | - | - |
| DiffSound | 400M | MBTG | CLIP | 65.68±1.58 | 55.91±1.75 | 7.39 | 2.57 |
| AUDIOGEN-base | 285M | - | T5-base | 70.85±1.06 | 63.23±1.65 | 2.84 | 2.14 |
| AUDIOGEN-base | 285M | Mix | T5-base | **71.68±1.89** | 66.01±1.79 | 3.13 | 2.09 |
| AUDIOGEN-large | 1B | Mix | T5-large | **71.85±1.07** | 68.73±1.61 | **1.82** | **1.69** |

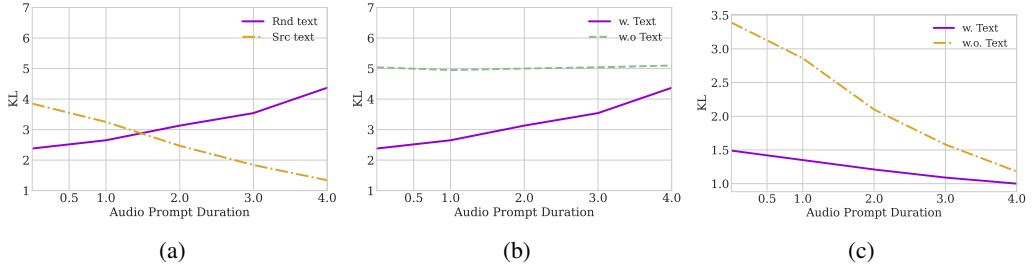

(a)  (b)  (c)

Figure 2: Audio continuation results. In (a) we compare model generations against audio corresponding to SRC and RND. In (b) we compare model generations with and without text conditioning against audio corresponding to RND text. In (c) we compare model generations with and without text conditioning against audio corresponding to SRC text. In all settings we report KL results. Specific details can be found on Section 4.2.

text input. We follow the MUSHRA protocol (Series, 2014), using both a hidden reference and a low anchor. For the overall quality test raters were asked to rate the perceptual quality of the provided samples in a range between 1 to 100. For the text relevance test, raters were asked to rate the match between audio and text on a scale between 1 to 100. Raters were recruited using the Amazon Mechanical Turk platform. We evaluate 100 randomly sampled files from the AudioCaps test set, where each sample was evaluated by at least 5 raters. We verified that the majority of the samples (85%) contain at least two audio concepts (e.g., "a dog barks while a bird chirps"). We use the CrowdMOS package[6] to filter noisy annotations and outliers. We remove annotators who did not listen to the full recordings, annotators who rate the reference recordings less then 85, and the rest of the recommended recipes from the CrowdMOS (Ribeiro et al., 2011). Participants in this study were paid at least the American minimum wage.

**Hyper-parameters.** We trained two sets of ALMs, one with 285M parameters (base) and the other with 1B parameters (large). In the smaller model we use a hidden-size of 768, 24 layers and 16 attention-heads, while for the large variant we use a hidden size 1280, 36 layers and 20 attention-heads. We use the Adam optimizer with a batch size of 256, a learning rate of 5e-4 and 3k steps of warm-up followed by inverse-square root decay. The small model was trained on 64 A100 GPUs for 200k steps (∼5 days) and the large model was trained on 128 A100 GPUs for 200k steps (∼1 week). For the small model we use T5-base and for the large model we use T5-large. For sampling, we employ top-$p$ (Holtzman et al., 2019) sampling with $p = 0.25$. For the CFG we use a $\gamma = 3.0$.

## 4.2 RESULTS

We start by comparing AUDIOGEN to DiffSound. Objective metrics are reported on the AudioCaps test set. We use the official pre-trained model provided by DiffSound authors [7]. The model was trained on AudioSet and fine-tuned on AudioCaps. Results are presented in Table 1.

---

[6]http://www.crowdmos.org/download/

[7]Pre-trained model can be found under https://github.com/yangdongchao/Text-to-sound-Synthesis

AUDIOGEN-base outperforms DiffSound considering all metrics while being smaller in terms of parameters count. As expected AUDIOGEN-large, significantly outperforms both DiffSound and AUDIOGEN-base. Notice, for AUDIOGEN-base, the model trained without mixing obtained superior FAD and comparable OVL to the model trained with mixing augmentations. However, when considering relevance to text, i.e., KL and REL. the model with mixing augmentations reaches better performance. This is especially noticeable for the REL. metric as it contains mostly complex compositions (see Table 1). We additionally compare AUDIOGEN-base to the same dataset setup used by DiffSound (i.e., training on AudioSet and AudioCaps). AUDIOGEN-base reached KL of 2.46 vs. 2.57 for DiffSound and FAD of 4.39 vs. 7.39 for DiffSound. These results suggest that AUDIOGENis still significantly superior to DiffSound.

Next, we experimented with pre-training the ALM component on only audio tokens without conditioning on text (learning an audio prior). We did not observe a gain in doing ALM pretraining. We hypothesize that is due to the pre-training process taking place on the same labeled data. Samples can be found under the following link: https://felixkreuk.github.io/audiogen.

**Audio Continuation.** Similarly to (Lakhotia et al., 2021), we first encode an audio prompt into a sequence of audio tokens, feed it into the ALM, and sample a continuation. Unlike previous work, we can additionally steer the generated audio towards textual captions. We evaluate our model given audio prompts of lengths [0.5s, 1s, 2s, 3s, 4s] and different combinations of text prompts: (i) no-text; (ii) text corresponding to the audio prompt (SRC); (iii) text taken from a randomly sampled audio file (RND). For each of the above settings we measure the KL between the generated audio and either the audio corresponding to the source text or the target text.

In Figure 2(a) we input the model with an audio prompt together with RNDas conditioning text. We evaluate KL between the output of the classification model using the generated audio against either the audio prompt or the audio corresponding to the RND. Results suggest that by using short audio prompts we could better steer the generation towards the conditioning text. In contrast, using long audio prompts leaves less room for textual guidance. Text and audio prompts have roughly the same impact at $\sim$1.5s. In Figure 2(b) we input the model with an audio prompt with and without RNDtext conditioning. We evaluate the KL between the output of the audio classification model using the generated audio against the audio corresponding to the RND. Although using longer audio prompts leaves less room for textual guidance, we still observe a gap between generations with and without text, even when using longer audio prompts ($\sim$4s). This suggests that text has a significant effect on the generation process. Lastly, In Figure 2(c) we condition the model on an audio prompt with and without SRCtext conditioning. We evaluate the KL between the output of the audio classification model using the generated audio against the input audio. Results suggest that when using short audio prompts, text has a major impact on the generated output. However as we feed longer sequences, the audio is sufficient to steer the generation towards the target concept classes with a minimal gain when adding text.

Full results together with FAD scores for all settings together with a visual depiction can be found in Table 4 on the supplementary 7.2.

### 4.3 ABLATION STUDY.

**The effect of classifier-free guidance scale.** As pointed out by Ho & Salimans (2021), the choice of the CFG scale offers a trade-off between sample diversity and quality with respect to the conditioning text. To gauge the effect of the $\gamma$ parameter in our setting on Figure 3 we report results for $\gamma \in \{1.0, 2.0, 3.0\}$. Notice, setting $\gamma = 1$ is equivalent to a vanilla sampling procedure (no CFG).

Removing the CFG results in poor performance compared to $\gamma > 1.0$. The FAD score reaches its' minimum at $\gamma = 3.0$, while the KL monotonically decreases and converges at $\gamma = 4.0$. This implies that using $\gamma = 3.0$ provides the best trade-off in the evaluated setting between quality and diversity.

**Multi-stream processing.** To better understand the benefit of using multiple streams over a single stream we optimized three different audio discrete representation models using Down Sampling Factors (DSF) of {x32, x64, x128} using {1, 2, 4} different codebooks respectively. For a fair comparison we kept all models at 2048 codes overall (e.g., 2 codebooks of size 1024 for the x64 model and 4 codebooks of size 512 for the x128 model).

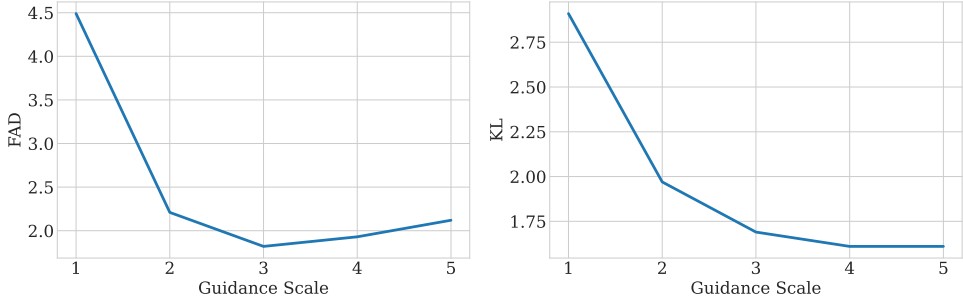

Figure 3: Results of FAD (left) and KL (right) as a function of the guidance scale.

Table 2: Multi-stream results: we report two sets of results encoding and generation. For the encoding metrics we report Bitrate (kbps), SI-SNR (dBs), and ViSQOL. For the generation scores we report FAD, KL and inference speed-up. We additionally include the number of streams and the Down Sampling Factor (DSF). Notice, the encoding metrics are the same for the large and base model as the same audio representation model was used.

| | # STREAMS | DSF | ENCODING | | | GENERATION | | |
| | | | Bitrate (kbps) | SI-SNR↑ | ViSQOL↑ | FAD↓ | KL↓ | SPEED-UP |
|---|---|---|---|---|---|---|---|---|
| base | 1 | x32 | 5.37 | 5.1 | 3.95 | 3.13 | 2.09 | x1.0 |
| | 2 | x64 | 4.88 | 4.5 | 3.94 | 10.35 | 2.17 | x2.0 |
| | 4 | x128 | 4.39 | 4.2 | 3.91 | 9.68 | 2.36 | x5.1 |
| large | 1 | x32 | 5.37 | 5.1 | 3.95 | 1.82 | 1.69 | x1.0 |
| | 2 | x64 | 4.88 | 4.5 | 3.94 | 6.89 | 1.86 | x2.3 |
| | 4 | x128 | 4.39 | 4.2 | 3.91 | 10.89 | 2.59 | x3.6 |

To assess the quality of the audio representation we report three encoding metrics namely Scale-Invariant Signal-to-Noise Ration (SI-SNR), ViSQOL (Chinen et al., 2020), and bitrate. These metrics characterize the audio representation only while ignoring the ALM. Note, both SI-SNR and ViSQOL are reference based metrics hence are computed between the audio reconstructed from the learned representation and the original audio sequence. Results are reported on Table 2. While a single stream encoder achieves better SI-SNR results, all representations are comparable in terms of estimated perceptual quality as measured by ViSQOL. While the total amount of codes is the same across all settings, the effective bitrate of the multi-stream models was degraded, leading to lower SI-SNR values.

Next, we report the FAD, KL and inference speed-up for the ALM on top of the learned representations. Results suggest that increasing the number of streams degrades the performance in both base and large models when compared to a single-stream. AUDIOGEN-base improves the KL score over DiffSound (2.57) while yielding higher FAD. AUDIOGEN-large with two streams improves over DiffSound in both FAD and KL (7.39 and 2.57 respectively). While showing inferior FAD and KL results when compared to the single-stream model, the multi-stream setting offers a trade-off between inference quality and speed.

## 5 LIMITATIONS

As we operate on audio tokens using relatively small down-sampling factor, the audio tokens sequences can be extremely long. This impose two main limitations: (i) modeling long range sequences; (ii) high inference time. In this work, we propose one possible relaxation to the first limitation, however such an approach comes at the cost of producing less quality audio samples. Such issues will become worse when considering high resolution audio samples, (e.g., sampling rate of 48kHz). Another limitation of the proposed approach relates to the audio compositions. Although, the mixing augmentation greatly improves the models ability to separate the sources and

create complex compositions, it is still lacks the understanding of temporal ordering in the scene, e.g., a dog is barking **then** a birds humming, vs. a dog is barking and a birds humming in the **background**. Lastly, as we omit most of the speech samples in our training set, the proposed approach often generates unintelligible speech. This can be mitigated by either using more speech data, better data augmentation recipes for speech, or by providing additional speech features. Another limitation of the datasets used is its diversity. The datasets were mainly collected from YouTube, in which specific demographic and geographic locations are more represent than others. This may create bias in the generated sample.

## 6    CONCLUSION

In this work, we propose a Transformer based generative model, named AUDIOGEN, which operates on a learnt discrete audio representation. Unlike previous work, we empirically demonstrate that auto-regressive models can generate high-quality audio samples conditionally or unconditionally. We show that on-the-fly text and audio mixing augmentations can improve model performance and provide an ablation study, analyzing the effect of CFG and multi-stream processing.

As for broader impacts, this work serves as the foundation for building better text-to-audio models. In addition, the proposed research could open up future directions involved with benchmarking, semantic audio editing, audio source separation from discrete units, etc.

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

## 7 APPENDIX

### 7.1 DATASETS

We use a set of several datasets: AudioSet (Gemmeke et al., 2017), BBC sound effects [8], Au-
dioCaps (Kim et al., 2019), Clotho v2 (Drossos et al., 2020), VGG-Sound (Chen et al., 2020),
FSD50K (Fonseca et al., 2021), Free To Use Sounds [9], Sonniss Game Effects [10], WeSoundEffects [11],
Paramount Motion - Odeon Cinematic Sound Effects [12]. All audio files were sampled at 16kHz.

Table 3: Datasets description. Duration is reported in hours for original audio, i.e., before pre-
processing.

| DATASET | TEXT CONDITIONING | DURATION (H) |
|---|---|---|
| AudioSet | tags | 5.42k |
| BBC | captions | 463 |
| AudioCaps | captions | 145 |
| Clotho v2 | captions | 37 |
| VGG-Sound | tags | 560 |
| FSD50K | tags & captions | 108 |
| Free To Use Sounds | tags & captions | 176 |
| Sonniss Game Effects | tags | 85 |
| WeSoundEffects | tags | 12 |
| Paramount Motion | tags | 20 |

### 7.2 ADDITIONAL RESULTS

We present a visual qualitative depiction of conditional and unconditional audio continuation in Fig-
ure 4. We use the following text as conditioning, "speech and a goat bleating", and the first second
of its corresponding audio as prompt. We mark the start of the generated segment using a dashed
white line. In Figure 4 (left) we visualize unconditional audio continuation (i.e., no text). In Fig-
ure 4 (right) we visualize audio continuation conditioned text. As the audio prompts contains only
human speech, the unconditional model generates a continuation that contains speech utterances, but
does not generate any goat sounds. In contrast, the model that is conditioned on text successfully
generates both human speech and goat sounds (left).

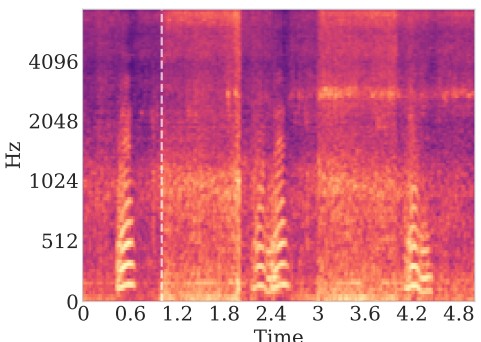 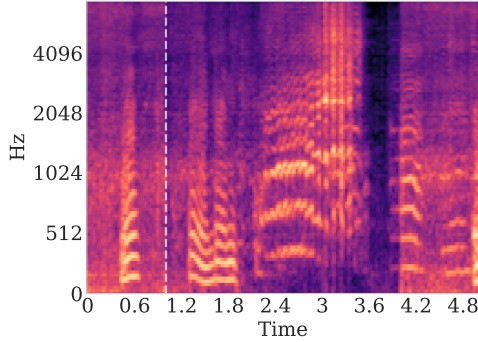

Figure 4: A visual example of the text guided audio continuation. We plot Mel-Spectrograms of
both audio continuation without text conditioning (left), and textually guided audio continuation
(right). Input text is: "speech and a goat bleating".

In table 4 we report the complete results presented in Figure 2.

---

[8]https://sound-effects.bbcrewind.co.uk/
[9]https://www.freetousesounds.com/all-in-one-bundle/
[10]https://sonniss.com/gameaudiogdc
[11]https://wesoundeffects.com/we-sound-effects-bundle-2020/
[12]https://www.paramountmotion.com/odeon-sound-effects

| TEXT | AUDIO PROMPT DURATION (SEC.) | FAD | KL w. RND | KL w. SRC |
|---|---|---|---|---|
| SRC | 0.5 | 1.97 | - | 1.49 |
| no text | 0.5 | 5.66 | 5.04 | 3.39 |
| RND | 0.5 | 3.08 | 2.38 | 3.85 |
| SRC | 1 | 1.96 | - | 1.35 |
| no text | 1 | 4.70 | 4.95 | 2.86 |
| RND | 1 | 3.17 | 2.65 | 3.25 |
| SRC | 2 | 1.92 | - | 1.21 |
| no text | 2 | 3.15 | 5.00 | 2.10 |
| RND | 2 | 2.92 | 3.13 | 2.47 |
| SRC | 3 | 1.90 | - | 1.09 |
| no text | 3 | 2.46 | 5.04 | 1.58 |
| RND | 3 | 2.58 | 3.54 | 1.84 |
| SRC | 4 | 1.97 | - | 1.00 |
| no text | 4 | 2.14 | 5.10 | 1.18 |
| RND | 4 | 2.29 | 4.37 | 1.34 |

Table 4: Full results of the FAD and KL metrics for all text condition and audio prompt settings

Finally, we analyze the effect of model sizes when considering text encoder and ALM, on the generated audio. In Table 5 we report KL and FAD scores for four different combinations: {T5-base, T5-large} × {ALM-base, ALM-large}. When using a larger T5 encoder we observe a big improvement in terms KL, and minor improvement in FAD. On the other hand, when using larger ALM, we see a similar improvement in terms or KL with a significantly bigger improvement in FAD. Using both T5-large and ALM-large yields the best results overall.

| T5 | ALM | KL | FAD |
|---|---|---|---|
| Base | Base | 2.09 | 3.13 |
| Base | Large | 1.92 | 2.27 |
| Large | Base | 1.91 | 3.03 |
| Large | Large | 1.69 | 1.82 |

Table 5: Ablation study. We report KL and FAD scores for four different text-encoder and ALM setups, considering base and large models.

