# OpenReview forum: "AudioGen: Textually Guided Audio Generation"
_ICLR.cc/2023/Conference — ICLR 2023 poster_

### Official Review · Reviewer_pGs2 · 2022-10-21

**Confidence:** 4
**Correctness:** 3
**Technical Novelty And Significance:** 3
**Empirical Novelty And Significance:** 3
**Recommendation:** 8

**Clarity, Quality, Novelty And Reproducibility:**

The text-to-audio field is still quite new, and this paper is a clear and novel step forward in that domain. For the most part, the paper is clear enough that it could be reproduced, though source code and pretrained models would help.

However, there are a couple sections that I found unclear:

**Section 3.2, Multi-stream audio inputs.**

It's not clear to me how this setup works. Do the multiple tokens predicted at the same time cover the same timestep? Is there any conditioning of the tokens on each other, or are they predicted independently? Is the RVQ implementation providing increasing levels of quality, as in Soundstream, or is it somehow related to decoding multiple timesteps at once?

I think some concrete examples and maybe a figure would help clarify this section.

**Section 4.2 Audio Continuation**

I found Figure 2 and its associated paragraph difficult to follow due to the combination of source/random text and audio and what they were being evaluated against. Also, I think there is a typo in the figure because there are 2 lines labeled "Rnd text". I think just being a little more explicit in the description would make things clear, describing exactly what was used for the audio prompt, the text conditioning, and the target/evaluation audio.

**Other comments**

Section 1, paragraph 2: Did you mean to say "astronaut riding a horse"?

Section 1, paragraph 3: When you that some compositions are unlikely to be in the training dataset, do you have any way of quantifying or testing that?

Section 1, paragraph 4: What size of pre-trained T5 did you use?

Section 3, paragraph 1: tokes -> tokens

Section 3.1: Just to clarify, does this mean that RVQ was not used other than in the multi-stream experiments? So a single token represents a single timestep? Could you also be more clear here about what the stride is? How much time does a token cover?

Section 3.1, Training Objective: I think it's worth discussing why the reconstruction loss is frequency magnitude only but the discriminator can also see phase.

Section 3.2: Instead of saying the architecture is like a decoder-only GPT-2, but with cross-attention added, wouldn't it be simpler to say it's an encoder-decoder Transformer with a frozen (T5) encoder?

Section 4.1 Dataset: Several of the quote marks are backwards.

Section 4.1 Dataset: Do you always train with input text of the form "dog bark park"? How does this work when the model is tested with phrases like "a dog barks while somebody plays the trumpet in a busy street”? Do you just rely on the T5 encoder to encode full sentences to the same representation as the sequence of individual words?

Section 4.1 Hyperparameters: Missing "hidden size" before "768".



**Strength And Weaknesses:**

Strengths:
- The outputs of the model are remarkable and clearly follow the text prompts.
- There is a clear improvement over previous work, with plenty of both quantitative and qualitative evaluations.
- Ablation studies help determine the relative benefits of classifier-free guidance, mixing, etc.

Weaknesses:
- Some sections of the paper were not clear or difficult to follow (see next section).
- Audio quality is still relatively low resolution (16 KHz).
- Generated speech is unintelligible.
- Exact timing of overlapping audio events is not controllable.

The weaknesses other than the first one are acknowledged in the paper.

**Summary Of The Paper:**

AudioGen is a text to audio model, focusing primarily on non-speech, non-music audio events. It supports rendering multiple overlapping events simultaneously depending on the given text description and can generalize to unseen text/audio pairs by using a pre-trained language model encoder (T5).

The paper does several ablation studies exploring the effects of classifier-free guidance, text/audio training data mixing, and multi-stream decoding.

**Summary Of The Review:**

This paper makes significant contributions to the relatively new field of text-to-audio. Overall the paper is clear and well written, but a few sections could be made more clear. I believe the paper will be influential in the field.

---

> ### Author Response · Authors · 2022-11-13
> **Author response to Reviewer pGs2**
>
> **Regarding unintelligible speech**, this is true and by design as we filter out most of the speech data. We additionally mention this in the limitations section.
>
> **Regarding controllability of the overlap of audio events**, we mention this in the Limitations section. We believe AudioGen is a major step towards textually guided semantic audio generation, and such controllability is an important next step in text-to-audio. We hope that such research directions will be explored by more members of the community.
>
> **Regarding Multi-stream inputs**, when using more than a single stream, each time-step contains multiple tokens (one for each stream). Hence, the audio LM receives as input multiple tokens at each time-step, and also predicts multiple tokens per time-step (such details can be found under Multi-Stream paragraph on the section 3.2). In the reported results each stream is predicted independently of other streams, we additionally explored conditioning streams on one another but found no significant gains. The RVQ implementation is very similar to the one in SoundStream where each codebook provides another level of granularity. We have clarified that in the manuscript.
>
> **Regarding Figure 2**, the goal behind this set of experiments was to evaluate the balance between text conditioning and audio conditioning. For clarification, we splitted Figure 2(a) into two sub-figures.
>
> In Figure 2 (a) we input the model with an audio prompt together with a conditioning text taken from a randomly sampled audio file (“rnd text”). We evaluate KL between the output of a state-of-the-art audio classification model using the model’s output against either the audio prompt or the audio corresponding to the “rnd text”. Results suggest that while using short audio prompts we could better steer the generation towards the conditioning text. In contrast, using longer audio prompts leaves less room for textual guidance. Text and audio prompts have roughly the same impact at ∼1.5s.
>
> In Figure 2 (b) we input the model with an audio prompt with and without text conditioning. The text used for conditioning was taken from a randomly sampled audio file (“rnd text”). We evaluate the KL between the output of a state-of-the-art audio classification model using the model’s output against the audio corresponding to the “rnd text”. Although using longer audio prompts leaves less room to steer the model using text, even when using longer audio prompts (~4s) we still observe a gap between generations with and without text. This suggests that text has a significant effect on the generation process.
>
> In Figure 2 (c) we condition the model on an audio prompt with and without a conditioning text corresponding to the input audio prompt. We evaluate the KL between the output of a state-of-the-art audio classification model using the model’s output against the input audio. Results suggest that when using short audio prompts, text has a major impact on the generated output. However as we feed longer sequences, the audio is sufficient to steer the generation towards the target concept classes with a minimal gain when adding text.
>
> We clarified that in the manuscript.
>
>
> **Regarding the astronaut and horse example**, correct, we fixed this in the manuscript.
>
> **Regarding the sizes of T5 models used**, the T5-base model is 220M parameters, the T5-large model is 770M parameters.
>
> **Regarding RVQ and multi-stream**, in the single-stream setting we set the RVQ module to contain a single codebook (equivalent to VQ). The downsampling factor is x32, resulting in each code representing 2ms of audio; these details were mentioned in the “Multi-stream audio inputs” paragraph in Section 3.1.
>
> **Regarding the training objective**, in preliminary results we found that using the complex value STFT for the discriminator provides better performance than magnitudes only. Regarding reconstruction losses, notice, the phase is contained in the time-domain signal which is reconstructed using the L1 loss.
>
> **Regarding the encoder-decoder description**, generally speaking we agree, we wanted to stress the difference between the text encoding model and the audio generation model (as often the encoder and decoder are mirror images of one another). Nevertheless we added such encoder-decoder description as an alternative view of our system in the manuscript.
>
> **Regarding text**, all text is pre-processed and passed through the T5 encoder during both training and inference.
>
> We addressed minor comments and typos in the manuscript.

---

### Official Review · Reviewer_Tiv8 · 2022-10-24

**Confidence:** 5
**Correctness:** 3
**Technical Novelty And Significance:** 3
**Empirical Novelty And Significance:** 4
**Recommendation:** 8

**Clarity, Quality, Novelty And Reproducibility:**

In general the paper is clearly written, with appropriate motivation, references, and equations. As mentioned in the summary, the novelty of the paper is primarily empirical, demonstrating the viability of text2audio.

 Below I point out some things and questions that could be addressed to strengthen the clarity and reproducibility of the paper:

* In the section on page 5, "Multi-stream audio inputs" I inferred from the text that each stream is sampled independently given the logits for a frame of audio, is that correct? If so, it might clarify to make that explicit.
* In describing the datasets, it could help to make a table to clarify the datasets used, their size pre and post filtering, and any other characteristics that vary between datasets such as their primary composition (speech, birds, etc.) and examples of the labels.
* Figure 2 is a bit confusing compared to the rest of the paper. Especially the graph on the right, where "Rnd text" is used for labels for two lines, but I believe in one, it refers to the conditioning and in the other it refers to the reference. frankly, there's not a whole lot in those figures that isn't presented clearer in table 3. I believe the authors are trying to highlight the Delta KL between rnd text and source text in the left, and text cond and no text cond in the right. Just plotting that value itself with a single line would make things clearer I think, or just adding it to table 3 and removing figure 2.
* Table 2 likewise is a bit redundant with the encoder info duplicated for both rows. Perhaps consider splitting into two tables? Or at least noting the redundancy in the caption as they are independent on the size of the decoder.
* The paragraph right before "Limitations" compares AudioGen in Table 2 to DiffSound. It would help to add DiffSound to table 2 to make the comparison easier.


Small typos:
* page 2: "generation in two axis" -> "generation in two axes"
* page 3: "audio tokes" -> "audio tokens"
* page 6: "we remove annotator" -> "we remove annotators"
* page 8: "three encoding mehtods" -> "three encoding metrics"
* page 9: "Limitation" -> "Limitations"
* page 15: Caption says the text is "a crowd applauds followed by a woman and a man speaking", but the text above says it should be "speech and a goat bleating" which looks correct given the spectrogram.

**Strength And Weaknesses:**

Strengths
* Strong empirical results. On both quantitative and qualitative metrics, this paper clearly demonstrates a "0-to-1" improvement on a new task of text2audio, even given the limitations of labeled data and long sequence lengths.
* Evaluation. Good choice of metrics for the experiments at hand, both quantitative and qualitative.
* Ablation studies motivate the choice of CFG scale and trade-offs of the multi-stream generation strategy.

Weaknesses
* The benefits/need for the mixing data augmentation strategy is not well supported by experiments. The results in Table 1 from including mixing augmentation are, well "mixed" at best. More importantly, an important baseline is missing. For a situation with multiple sources, it would be good to compare to the baseline of generating several audio clips from a model trained without augmentation, and then mixing those outputs posthoc. Clearly this approach has the downside of taking longer to generate, but can handle arbitrary combinations of sources, regardless of the training. It also exposes a shortcoming of the current data augmentation strategy, as it ignores the fact that real audio often has many correlations between sound sources, including acoustic environment, background noise, and interactions in the natural scene.
* The comparisons with model size in Table 1 could also benefit from clarifying what improvements are due to increasing the size of the decoder (ALM) vs. increasing the size of the encoder (T5). Currently they are both scaled together, and an ablation experiment could help to demonstrate where the benefits emerge.
* It is unclear if DiffSound was retrained on the same datasets as AudioGen for the comparisons in Table 1. If the model is indeed trained on different datasets (using a pretrained checkpoint) then the comparison is less useful as a comparison of model architectures and needs to be highlighted as such. AudioGen is still an impressive empirical feat, but it would then be unclear how much is due to the model architecture vs. the datasets, and the claim couldn't be made that the AudioGen architecture outperforms the DiffSound architecture in this case.
* The limitations section doesn't sufficiently address limitations of the datasets. Datasets with fine-grained multi-labels would be a better approach at capturing true scene dependencies than the linear augmentation. Further, the model will be significantly biased by the distribution of the training data in ways that cut across demographics and geography. As the quality of these models improve, this can present bias and representation challenges in a similar manner to text2image models currently, and it's important to note that in the paper.

**Summary Of The Paper:**

The combine existing methods (SoundStream audio codec, Transformer LMs) and datasets in a new way to demonstrate significantly higher quality generation of text-conditional environmental sounds than previous state of the art. Architecture and loss functions are largely the same to previous papers, but the extension to text-to-audio makes this paper a significant contribution to the community and a demonstration of the potential for follow-up research.

The authors also explore a variety of extensions to the task at hand (augmenting data by mixing audio clips/captions, speeding up inference through predicting multiple independent streams of audio tokens. The experiments are clearly laid out provide both quantitative and qualitative metrics to support the improvement over a baseline method (DiffSound) and the increase of quality with increasing model size. Technical limitations of the approach are appropriately explained.

**Summary Of The Review:**

This paper presents an impressive empirical demonstration of text2audio using existing techniques. Datasets are carefully curated and methods selected to enable this demonstration, and it is of high significance to the ML community as it points towards the potential of pushing these techniques further, and the limitations of current datasets and methods.

In general, the experiments are carefully performed, with good metrics and ablations. Several important questions remain to ensure that the comparisons between models are fair, the value of techniques such as mixing data augmentation are justified, and the relative importance of scaling encoders / decoders are understood. Some discussion of future ethical implications would also help the limitations section of the paper.

---

> ### Author Response · Authors · 2022-11-13
> **Author response to Reviewer Tiv8 (part 1)**
>
> **Regarding the suggested baseline (post-hoc mixing)**: Thanks for the suggestion. Although such a baseline might seem “simple” at first, one must first take into consideration multiple factors: 1) text preprocessing to extract entities using one of the existing NLP techniques; 2) After the entities were extracted, we are left with words that might describe either the entities (“quietly” / “loudly” / “fast”) or the relationship between them (“in the background”). This would also require understanding how such words relate to the extracted entities (e.g. does the word “quietly” relate to “dog” or “man”); 3) After each audio source was generated, one must then decide how these sources will be mixed together (e.g., what SNR does “background” / “foreground” mean? What is the temporal alignment? etc.). Furthermore, some descriptions such as “whistling in the wind” correspond to the recording conditions and not necessarily to the composition.
> Due to all of the above, although such a baseline is an interesting and intuitive research direction it is not trivial and requires careful consideration, tuning, and evaluations of multiple components in a cascaded pipeline. Hence, we believe it is out of the scope of this submission.
>
> **Regarding results in Table 1:** We conduct the experiments requested by the reviewer:
>
> |   T5  | ALM   |  KL  | FAD  |
> |:-----:|-------| :--: | -----|
> |  Base | Base  | 2.09 | 3.13 |
> | Base  | Large | 1.92 | 2.27 |
> | Large | Base  | 1.91 | 3.03 |
> | Large | Large | 1.69 | 1.82 |
>
> When using a larger T5 encoder we observe a big improvement in terms KL, and minor improvement in FAD. On the other hand, when using larger ALM, we see a similar improvement in terms or KL with a significantly bigger improvement in FAD. Using both T5-large and ALM-large yields the best results overall. We included these results in the appendix.
>
> **Regarding the need for mixing augmentations**: we do observe a significant improvement in the results. This is especially notable when considering the subjective text-relevance and KL scores.
>
> **Regarding correlations of sound sources in real audio**, indeed natural audio has many correlations in its sound sources. Nevertheless, we argue that learning and relying on these correlations would lead to a weaker model. For example, the training data does not contain compositions of “cat” and “keyboard typing”. We would therefore like to create unseen compositions such that the model would be minimally biased towards the spurious correlations of the training data.
>
> **Regarding the comparison to DiffSound**, The dataset used to train our model (1.2M samples, 4k hours) is smaller than the dataset used to train DiffSound (1.8M samples, 5.8k hours). Having said that, we agree that for a better comparison we should also evaluate AudioGen on the same dataset used to train DiffSound. To that end, we provide the following FAD and KL metrics: AudioGen-base reached a FAD score of 4.39 and a KL score of 2.46.
> The results suggest that AudioGen-base still outperforms DiffSound when evaluated on the same data. Results are also updated in the manuscript.
>
> **Regarding the limitations of the datasets**, we added a clarification in the manuscript.
>
> **Regarding multi-stream processing**, we experimented with both independently sampling codes and condition the sampling on previous codes. We did not observe a significant gain in performance. We clarified that in the paper.
>
> **Regarding dataset descriptions**, we included a table with the number of hours and text conditioning in the appendix (see Table 3).

---

> > ### Author Response · Authors · 2022-11-13
> > **Author response to Reviewer Tiv8 (part 2)**
> >
> > **Regarding Figure 2**, the goal behind this set of experiments was to evaluate the balance between text conditioning and audio conditioning. For clarification, we splitted Figure 2(a) into two sub-figures.
> >
> > In Figure 2 (a) we input the model with an audio prompt together with a conditioning text taken from a randomly sampled audio file (“rnd text”). We evaluate KL between the output of a state-of-the-art audio classification model using the model’s output against either the audio prompt or the audio corresponding to the “rnd text”. Results suggest that while using short audio prompts we could better steer the generation towards the conditioning text. In contrast, using longer audio prompts leaves less room for textual guidance. Text and audio prompts have roughly the same impact at ∼1.5s.
> >
> > In Figure 2 (b) we input the model with an audio prompt with and without text conditioning. The text used for conditioning was taken from a randomly sampled audio file (“rnd text”). We evaluate the KL between the output of a state-of-the-art audio classification model using the model’s output against the audio corresponding to the “rnd text”. Although using longer audio prompts leaves less room to steer the model using text, even when using longer audio prompts (~4s) we still observe a gap between generations with and without text. This suggests that text has a significant effect on the generation process.
> >
> > In Figure 2 (c) we condition the model on an audio prompt with and without a conditioning text corresponding to the input audio prompt. We evaluate the KL between the output of a state-of-the-art audio classification model using the model’s output against the input audio. Results suggest that when using short audio prompts, text has a major impact on the generated output. However as we feed longer sequences, the audio is sufficient to steer the generation towards the target concept classes with a minimal gain when adding text.
> >
> > We clarified that in the manuscript.
> >
> > **Regarding the results in Table 2:** We updated the caption of Table 2 to point out the encoding metrics are the same for both base and large models as we use the same encoding model.
> >
> > We addressed all the minor comments in the manuscript.

---

### Official Review · Reviewer_iK2a · 2022-10-25

**Confidence:** 4
**Correctness:** 3
**Technical Novelty And Significance:** 3
**Empirical Novelty And Significance:** 3
**Recommendation:** 8

**Clarity, Quality, Novelty And Reproducibility:**

-Some aspects regarding the experimental details are not super clear to me. For instance, as I wrote above I am not sure about under which exact conditions did you compare with Diffsound. Also it is not indicated how long does the training takes with the indicated number of GPUs.

-The proposed architecture is not particularly novel from a technical standpoint, but the curation of several different aspects (dataset, design of the architecture, delivery of the results) make this work good quality.

-Reproducibility is extremely difficult as the number of GPUs used in this dataset is very large. Also the code is not available.

**Details Of Ethics Concerns:**

Unfortunately successful generative models come with the danger to be used for malicious intents. This is not something particularly specific to this work though.

**Strength And Weaknesses:**

Strengths:
-The results sound impressive.
-Methodology seems clear.
-The paper is relatively well written.

Weaknesses:
-The employed model seems to be very large, and the employed computational resources are enormous.
-I am not sure if the comparison with diffsound is exactly fair. I am sorry if I am missing something, but it doesn't seems to be indicated how did you get the results with Diffsound. Also, a comparison on how much data each model uses. From what I understand you use more data than diffsound to train your model. Ideally it would be nicer to train the architecture of diffsound on the dataset you have curated. This would help to ascertain if your proposed architecture contributes more significantly than the size of the dataset. I am sorry if you have done this experiment, however from the manuscript I do not see it.



**Summary Of The Paper:**

This paper proposes a solution to generate audio from text prompts. The proposed solution uses a Vector-Quantized VAE, and in the latent space a language model is employed.

**Summary Of The Review:**

There is some unclear aspects I noted above. However I think this is a good quality work, and therefore it is above the acceptance threshold.

---

> ### Author Response · Authors · 2022-11-13
> **Author response to Reviewer iK2a**
>
> **Regarding the comparison to DiffSound**, we would like to note that AudioGen-base (285M parameters) is smaller than DiffSound (400M parameters). Moreover, the dataset used to train our model (1.2M samples, 4k hours) is smaller than the dataset used to train DiffSound (1.8M samples, 5.8k hours). Having said that, we agree that for a better comparison we should also evaluate AudioGen on the same dataset used to train DiffSound. To that end, we provide the following FAD and KL metrics: AudioGen-base reached a FAD score of 4.39 and a KL score of 2.46.
> The results suggest that AudioGen-base still outperforms DiffSound when evaluated on the same data. Results are also updated in the manuscript.
>
> **Regarding the experimental details**, we used the official DiffSound implementation and pre-trained model to generate audio conditioned on the AudioCaps test set. The training time for AudioGen-base was ~5 days using 64 GPUs and for AudioGen-large was ~1 week using 128 GPUs. We added the above details in the manuscript.
>
> **Regarding the code**, we will share the code and pre-trained models upon publication.

---

### Official Review · Reviewer_J2ki · 2022-10-27

**Confidence:** 5
**Correctness:** 3
**Technical Novelty And Significance:** 3
**Empirical Novelty And Significance:** 4
**Recommendation:** 8

**Clarity, Quality, Novelty And Reproducibility:**

- Clarity is high. Clear writing about what was done.
- Quality is high. The proposed system is well designed with state-of-the-art architectures and sensible, intuitive, and efficient idea of using the available datasets.
- Novelty is medium-high (or high enough)
- Reproducibility is high. In practice though, with the current computational cost, it is not easy to actually reproduce this work.

**Strength And Weaknesses:**

Strength:
- Clever use of available datasets
- Well designed and executed experiment that shows the system works well.
- Limitation of the proposed system is given

**Summary Of The Paper:**

The authors present a system named AudioGen that can generate audio given text description of sound. The system is trained using a set of many audio datasets. Unlike (image, text) pairs, the size of (audio, text) pairs is not large enough, at least in the public domain. This limitation is tackled by using audio tag labels. Although this means what AudioGen does is rather a tag-based generation than text description, this is probably the best output using the currently available dataset. The loss is also carefully chosen, which seems to help.

**Summary Of The Review:**

I believe the paper is good enough to be accepted overall. I'll only describe some issues here.
- After the text processing, is it correct to call them "sentences"? I find this a bit misleading. Although a description-based system does not need to fully reflect what is written in the description (it is probably not possible), the proposed system seems rather (open vocabulary) tag-based generation than sentence-based one.
- Very minor issue, but In 4.3, last paragraph: "AudioGen improves the KL score over.. higher FAD" → it'd be easier to follow if the scores of DiffSound is presented again in this sentence.
- In 5. Limitation, I don't think such a system is supposed to generate intelligible speech (it's not a TTS system.)
- It'll be useful if the text preprocessing code is shared somewhere. Not every reader is knowledgeable on text processing.

---

> ### Author Response · Authors · 2022-11-13
> **Author response to Reviewer J2ki**
>
> **Regarding text processing**, we agree that the processed text might not look like a natural sentence. However, it is also not completely equivalent to using tags only. For example, descriptive phrases such as “X with Y **in the background” will contain more than just the X and Y tags, namely, it will also contain descriptive words like “background”. We changed the phrasing to reflect that and named them to “pseduo-sentences”.
>
> **Regarding the code for text processing**, we will share the code upon publication.
>
> **Regarding adding the scores of DiffSound**, we added them to the paper.

---

### Public Comment · ~Zeqian_Ju1 · 2022-11-13
**Questions about muti stream & demos**

The results are quite impressive. I think the generated samples even have better quality than Ground Truth.  I would like to check the following questions：
1. As shown in Table 2, the single stream setting can achieve the best performance in all metrics except for speed.  And I guess that the samples on demo page are generated under single stream setting if not annotated.  Is it right?

2. Are Ground Truth on demo page reconstructed from tokens, or raw waves from datasets?

3. In ``Multi-stream modelling`` section on demo page, I notice that the generated samples sound much more 'intense' and 'speedy', as the stream number grows. Is it a common phenomenon, and why?

Thanks a lot for your response!

---

> ### Author Response · Authors · 2022-11-13
> **Answer to Zeqian Ju**
>
> **Regarding Table 2:** Yes, the samples shared on the demo page are single stream unless stated otherwise. Generally speaking as you rightfully mentioned, the single stream setup got the best results, however, this is at the expense of slower inference time.
>
> **Regarding ground truth samples on the demo page:** These are the original raw-waveform samples from AudioCaps, sampled at 16kHz.
>
>  **Regarding ground truth samples on the demo page:** We did not observe that the multi-stream samples are more “intense” or “speedy”. We did notice that the multi-stream samples are noisier than the single stream ones, which might create the illusion of intensity.

---

### Author Response · Authors · 2022-11-13
**Thank you**

We would like to thank the reviewers for taking the time to review our paper, and provide helpful comments. We are glad to see the reviewers found the paper to be valuable.
We have updated the paper with the requested changes, more details can be found below.

---

### Decision · Program_Chairs · 2023-01-20

**Decision:**

Accept: poster

**Justification For Why Not Higher Score:**

The novelty of the paper mainly lies in developing a high quality solution to the environment sound generation.

**Justification For Why Not Lower Score:**

 The study and the presentation are solid. The evaluation, ablation studies and discussions on limitations are sound.

**Metareview: Summary, Strengths And Weaknesses:**

Summary: the paper combines existing methods (SoundStream, Transformer LM) and datasets in a new way to demonstrate significantly higher quality generation of text-conditioned environmental sounds.

Strengths: The empirical results are strong. The presentation of the paper is clear and ablation studies are solid.

Weaknesses: The novelty of the paper is primarily empirical.

**Note From Pc:**

if the above contains the word "oral" or "spotlight" please see: "oral" presentation means -> notable-top-5% and "spotlight" means -> notable-top-25%. As stated in our emails, we are disassociating presentation type from AC recommendations